# Random Fourier Features-Based Deep Learning Improvement with Class Activation Interpretability for Nerve Structure Segmentation

**DOI:** 10.3390/s21227741

**Published:** 2021-11-20

**Authors:** Cristian Alfonso Jimenez-Castaño, Andrés Marino Álvarez-Meza, Oscar David Aguirre-Ospina, David Augusto Cárdenas-Peña, Álvaro Angel Orozco-Gutiérrez

**Affiliations:** 1Automatic Research Group, Universidad Tecnológica de Pereira, Pereira 660003, Colombia; dcardenasp@utp.edu.co (D.A.C.-P.); aaog@utp.edu.co (Á.A.O.-G.); 2Signal Processing and Recognition Group, Universidad Nacional de Colombia, Manizales 170003, Colombia; amalvarezme@unal.edu.co; 3Medicina Hospitalaria, Servicios Especiales de Salud (SES) Hospital de Caldas, Manizales 170003, Colombia; odaguirre@ses.com.co

**Keywords:** nerve structure segmentation, ultrasound images, deep learning, random Fourier features, class activation mapping

## Abstract

Peripheral nerve blocking (PNB) is a standard procedure to support regional anesthesia. Still, correct localization of the nerve’s structure is needed to avoid adverse effects; thereby, ultrasound images are used as an aid approach. In addition, image-based automatic nerve segmentation from deep learning methods has been proposed to mitigate attenuation and speckle noise ultrasonography issues. Notwithstanding, complex architectures highlight the region of interest lacking suitable data interpretability concerning the learned features from raw instances. Here, a kernel-based deep learning enhancement is introduced for nerve structure segmentation. In a nutshell, a random Fourier features-based approach was utilized to complement three well-known semantic segmentation architectures, e.g., fully convolutional network, U-net, and ResUnet. Moreover, two ultrasound image datasets for PNB were tested. Obtained results show that our kernel-based approach provides a better generalization capability from image segmentation-based assessments on different nerve structures. Further, for data interpretability, a semantic segmentation extension of the GradCam++ for class-activation mapping was used to reveal relevant learned features separating between nerve and background. Thus, our proposal favors both straightforward (shallow) and complex architectures (deeper neural networks).

## 1. Introduction

Recently, regional procedures have been arisen as an attractive alternative for general anesthesia in the context of medical surgeries to enhance post-operative mobility and reduce mortality and morbidity [1]. In this sense, peripheral nerve blocking (PNB) is a widely used method that involves the administration of an anesthetic substance in the area surrounding a nerve structure to block the transmission of nociceptive information [2]. Nevertheless, the success of PNB depends on a nerve structure’s precise localization, avoiding adverse effects such as neurological damage or intoxication due to the flow of the anesthetic into the bloodstream [3]. Concerning this, ultrasonography has been used to support PNB. This technique aims to improve targeting accuracy enabling real-time visualization of the nerve at low cost, while also being non-invasive and using no radiation [4].

Conventional 2D ultrasound images carry different challenges such as attenuation, artifacts, and speckle noise-based disturbances, which make the nerve location by visual inspection a difficult task [5]. Therefore, image-based automatic nerve segmentation systems have been proposed to assist the anesthesiologist in locating nerves during PBN [6]. Indeed, nerve segmentation from ultrasound has been widely studied in recent years. Among the works proposed, feature engineering-based techniques employ wavelet transform, standard deviation, and entropy-based super-pixel representations. Further, the predefined features are used to feed a classifier for nerve segmentation [7,8]. Nevertheless, hand-computed features can yield poor segmentation results, not to mention their high computational burden [9,10].

Conversely, deep learning-based methods have been introduced to compute the representation space and the classifier as a whole approach within a semantic segmentation strategy. Remarkably, deep learning has an advantage over classical machine learning algorithms, especially for image processing tasks, mainly because of its “automatic” feature learning capability on large datasets, high performance, and faster inference [11]. For instance, in [12] the authors proposed a fully convolutional network (FCN) approach that replaces the well-known fully connected layer with a convolutional architecture to predict a class label to each image’s pixel. However, FCN applies lower-level image features to infer higher-level content [13]. Next, in [14], the authors present a U-net architecture for biomedical image segmentation, which extracts the image features of each layer using down-sampling and up-sampling schemes to segment the target. The latter helps to find insight by exploring advanced image attributes but, at the same time, causes a reduction in the size of the feature map. If the goal mask is covered by a black shadow or is not apparent, the U-net network cannot achieve an accurate segmentation [15].

Further, in [16], the ResNet and U-Net approaches are combined to build the so-called residual neural network and U-net (ResUnet). Such an algorithm aims to extract deeper information from input images. Indeed, some ResUnet variants include batch normalization and weight initialization approaches [13], achieving relevant semantic segmentation results but holding a more complex architecture than FCN and U-Net models [17]. Furthermore, several authors have proposed deep learning variants devoted to semantic segmentation from the previously mentioned models in the last years. For instance, in [18] the authors introduced an automatic nerve structure segmentation methodology founded on U-net. In turn, linear Gabor binary patterns (LGBP) and a deep learning strategy are used to segment nerves from ultrasound images [19]. In [20] a median filter is used to reduce the speckle noise artifacts. Then, dense atrous convolution (DAC) with residual multi-kernel pooling (RMP) is utilized to enhance the ResUnet performance. Still, complex architectures are required to improve the segmentation performance, not to mention the lack of a suitable data interpretability [6,21,22].

On the other hand, kernel methods are known for their representation benefits among classical machine learning techniques. Mainly, these techniques have two main advantages: convex optimization and theoretical assurances for model generalization [23]. Nevertheless, straightforward kernel-based algorithms demand a high computational burden for computing the Gram matrix and provide low efficiency compared to deep learning models [24]. In this sense, deep learning approaches perform a high-level abstraction from input space, bypassing the representation through layers to provide local perturbation invariance and suitable prediction performances [25].

Intending to exploit both kernel and deep learning benefits, some authors aim to combine them from two main perspectives [26]: deep kernel learning and explicit mapping kernel approximation. The former focused on learning a kernel function from data using neural networks [27,28,29,30,31,32]. For example, in [29], authors proposed a convolutional kernel network (CKN) as an unsupervised approach to approximate kernel mappings. Next, in [31], a random Fourier feature transform is employed to embed deep architectures and convolutional filters into kernel-based schemes. Though deep kernels obtain good learning performance using the implicit stochastic mini batch-based regularizer, model overfitting, and unstable training persist in most of the cases [33].

In turn, approaches based on explicit kernel approximation seek to estimate the non-linear mapping directly. The fundamental work in [34] introduces the random Fourier features (RFF) estimator founded on Bochner’s theorem for stationary kernels [35]. The RFF approach tackles the issues of significant computational and storage cost of kernel matrices [36]. In addition, some RFF variants (most of them approximating a Gaussian-based mapping) have been proposed to improve the computational cost and learning performance. For instance, the fast-food algorithm employs structured matrices to reduce the RFF’s computational time from $O(Q^{\prime }P^{\prime })$ to $O(Q^{\prime }\log P^{\prime })$, where $Q^{\prime }$ represents the number of random features to represent and $P^{\prime }$ is the input data dimensionality [37]. After that, the orthogonal random features (ORF) approach computes the RFF’s Gaussian matrix with an orthogonal-based procedure [38]. Likewise, the structured orthogonal random features (SORF) method extends the ORF technique using a normalized Walsh–Hadamard matrix [38]. The butterfly-based quadrature (BQ) variant handles butterfly matrices to improve the system performance [39]. Notwithstanding, RFF-based techniques depend upon a trade-off between the system accuracy and computational burden [40].

This work proposes a kernel-based strategy to support nerve structure segmentation from ultrasound images using deep learning. Concerning this, an RFF-based approach is employed to approximate a Gaussian kernel implicit mapping within three well-known architectures for image-based semantic segmentation. In particular, the FCN [12], Unet [14,41], and ResUnet [16] are studied. Our RFF-based improvement aims to provide a better generalization capability for ultrasound image-based nerve segmentation using straightforward and complex architectures. For concrete testing, we coupled an RFF layer within the bottleneck end for Unet and ResUnet architectures; meanwhile, the last pooling layer was used to locate our kernel enhancement in FCN. Moreover, two ultrasound image datasets were tested. The former belongs to the Universidad Tecnológica de Pereira and the Santa Mónica Hospital, Dosquebradas, Colombia; holding ultrasound images of sciatic, ulnar, median, and femoral nerves. The latter is a Kaggle Competition dataset [42], gathering ultrasound images of the brachial plexus (BP). For data interpretability, a semantic segmentation extension of the gradient class activation mapping (GradCAM++) strategy was applied [43], which aims to visually test the deep learning model’s ability to learn relevant features separating between nerve and background. Specifically, a Grad-CAM++ extension of the seminal work in [44] for semantic segmentation was proposed to capture the entire object completeness. Then, an explanation map-based quantitative assessment was carried out for relevance analysis. Obtained results prove that our RFF-based improvement facilitates the discrimination between nerve structure and background with preserved data interpretability concerning the highlighted image regions.

The remainder of this paper is organized as follows. Section 2 depicts the materials and methods. Section 3 and Section 4 present the experimental setup and the results obtained. Finally, Section 5 shows the concluding remarks.

## 2. Materials and Methods

### 2.1. Deep-Learning-Based Semantic Segmentation Fundamentals

Let ${\left\{I_{n}\in R^{R\times C},\;M_{n}\in {\left\{0,1\right\}}^{R\times C}\right\}}_{n=1}^{N}$ be an input–output set holding *N* labeled images, where $I_{n}$ is the *n*-th image with *R* rows and *C* columns. The mask $M_{n}$ encodes the one-hot membership of each pixel on $I_{n}$ to the target class. For simplicity, gray-scale images and a binary segmentation problem are considered, e.g., background vs. object of interest.

A deep learning architecture for semantic segmentation often includes a stack of convolutional layers fed by the input images to predict each pixel label by exploiting the local spatial correlations. Thereby, let ${\left\{W_{l}\in R^{P_{l}\times P_{l}\times D_{l}}\right\}}_{l=1}^{L}$ be a set of convolutional layers, where $P_{l}$ and $D_{l}$ denote the *l*-th layer size and the the number of filters, respectively (*L* holds the number of convolutional layers). Given an input image I, a prediction mask $\hat{M}\in {\left[0,1\right]}^{R\times C}$ can be computed as:(1)$$\hat{M}=\left(\varphi_{L}\circ \ldots \circ \varphi_{1}\right)\left(I\right),$$
where $F_{l}=\varphi_{l}\left(F_{l-1}\right)=\nu_{l}\left(W_{l}⊗F_{l-1}+b_{l}\right)\in R^{R_{l}\times C_{l}\times D_{l}}$ is a tensor holding $D_{l}$ feature maps at the *l*-th layer, $\varphi_{l}:R^{R_{l-1}\times C_{l-1}\times D_{l-1}}\to R^{R_{l}\times C_{l}\times D_{l}}$ is a representation learning function, $b_{l}\in R^{D_{l}}$ is a bias vector, and $\nu_{l}\left(\cdot \right)$ is a non-linear activation function, i.e., rectified linear unit defined as ReLU(x)=max(0,x). Notation ∘ stands for function composition and ⊗ for image-based convolution. Note that $F_{0}=I$ and $F_{L}=\hat{M}$, where $\nu_{L}\left(\cdot \right)$ can be fixed as a sigmoid or softmax function for bi-class and multiclass segmentation, respectively. In turn, the prediction accuracy relies on the parameter set $\theta ={\left\{W_{l},b_{l}\right\}}_{l=1}^{L}$, yielding:(2)$$\theta^{*}=\arg\min_{\theta }E\left\{L\left(M_{n},{\hat{M}}_{n}|\theta \right):\forall n\in \left\{1,2,\ldots ,N\right\}\right\},$$
where $L:{\left\{0,1\right\}}^{R\times C}\times {\left[0,1\right]}^{R\times C}\to R$ is a given loss function and E{·} stands for the expected value operator. The optimization problem in Equation (2) can be solved through mini-batch based gradient descend using back-propagation and automatic differentiation [45].

Concerning this, the representation learning stage depicted on the composition function in Equation (1) can be built from several deep learning architectures. Consequently, the three most relevant approaches devoted to image-based semantic segmentation are briefly described:–*Fully convolutional network* (FCN) [12]: It is known as the fundamental semantic segmentation architecture that avoids computational redundancy and replaces fully connected layers with convolutional ones. FCN is based on the well-known “very deep convolutional network for large-scale image recognition model” (also known as the VGG-16 algorithm) [46].–*U-net* [14]: This approach aims to extract low-level features while preserving high-level semantic information. Moreover, the U-net algorithm pretends to relieve training problems related to a limited number of samples [47]. Remarkably, the U-net’s architecture includes an encoder and decoder stage, and is a U-shaped network.–*Residual network and U-net* (ResUnet) [16]: This approach enhances the U-net algorithm including residual blocks. Thereby, residual learning is employed to boost the model layers as residual functions referenced to the inputs, instead of learning unreferenced mappings; that is, the enhanced feature maps can be rewritten as $F_{l}=\varphi_{l}\left(F_{l-1}\right)+F_{l-1}$ [48]. Then, the ResUnet combines low and high-level features, favors the network optimization, and includes a deeper representation learning stage than U-net and FCN approaches.

### 2.2. Random Fourier Features Approximating Kernel Mappings

To improve the generalization capability of deep learning approaches for semantic segmentation, we propose to include a kernel mapping-based layer within the network architecture. For such a purpose, let $x,x^{\prime }\in R^{P}$ be a pair of samples from a real random variable on *P* dimensions. The well-known kernel trick indirectly computes the inner product between implicitly generated features from any pair $x,x^{\prime }$ using a kernel function $\kappa :R^{P}\times R^{P}\to R$, so that $\kappa \;\left(x,\;x^{\prime }\right)=\langle \phi \left(x\right),\phi \left(x^{\prime }\right)\rangle$, where $\phi :R^{P}\to H$ defines an implicit mapping to an “infinite-dimensional” Hilbert space H.

Due to the untractable mapping, kernel approaches require high computational and storage costs for large training sets. The random Fourier features (RFF) method lightens the computational burden by taking advantage of Bochner’s theorem for shift-invariant kernels, e.g., $\kappa \left(x-x^{\prime }\right)=\kappa \;\left(x,\;x^{\prime }\right)$ [34]. Namely, a function $\kappa (x-x^{\prime })$ is positive definite if and only if its Fourier transform is related to a non-negative measure, as follows:(3)$$\kappa \left(x-x^{\prime }\right)=\int_{R^{P}}p\left(\omega \right)e^{j\omega^{⊤}\left(x-x^{\prime }\right)}d\omega =E\left\{\zeta_{\omega }\left(x\right)\zeta_{\omega }{\left(x^{\prime }\right)}^{*}\right\},$$
where $\zeta_{\omega }\left(x\right)=e^{j\omega^{⊤}x}$, so that $\zeta_{\omega }\left(x\right)\zeta_{\omega }{\left(x^{\prime }\right)}^{*}$ is an unbiased estimate of $\kappa (x-x^{\prime })$ when $\omega \in R^{P}$ is drawn from p(ω) and ${}^{*}$ stands for the complex conjugate.

Moreover, the probability p(ω) and the kernel function κ(·) are real, then the integral in Equation (3) converges by replacing the exponential with a cosine. So, a real-valued mapping that satisfies the condition $E\left\{\tilde{\varphi }\left(x\right)\tilde{\varphi }{\left(x^{\prime }\right)}^{*}\right\}=\kappa \left(x-x^{\prime }\right)$ can be obtained fixing $\tilde{\varphi }\left(x\right)=\sqrt{2}\cos\left(\omega^{⊤}x+b\right),$ when ω∼p(ω) and b∼U(0,2π).

Since the expected value of $\tilde{\varphi }\left(x\right)\tilde{\varphi }{\left(x^{\prime }\right)}^{*}$ converges to $\kappa (x-x^{\prime })$, the estimator’s variance can be reduced by concatenating *Q* randomly chosen mappings (normalized by $\sqrt{Q}$); then the following approximation arises [49]:(4)$$\tilde{\varphi }\left(x\right)\approx \hat{\varphi }\left(x\right)=\sqrt{\frac{2}{Q}}{\left[\cos(\omega_{q}^{⊤}x+b_{q})\right]}_{q=1}^{Q},$$
where $\hat{\varphi }:R^{P}\to R^{Q}.$ Overall, the Gaussian kernel is preferred because of its universal approximating property and mathematical tractability [50]. Then, for $\kappa (x-x^{\prime })=$
$\exp(-{∥x-x^{\prime }∥}_{2}^{2}/2\sigma^{2})$, being $\sigma^{2}\in R^{+}$ a given bandwidth, its Fourier transform yields to $p\left(\omega \right)=N\left(0,\sigma^{2}\overset{˘}{I}\right)$, where 0 and $\overset{˘}{I}$ are an all-zero vector and the identity matrix of proper size.

Afterward, to provide a better generalization capability founded on kernel mappings, an RFF-based layer from Equation (4) can be used to enhance the semantic segmentation architectures exposed in Section 2.1, by adding $\hat{\varphi }\left(\cdot \right)$ to the function composition approach in Equation (1). In particular, we propose to add the RFF layer after the last pooling in FCN (see Figure 1). Similary, the RFF mapping is added after the bottleneck end for U-net and ResUnet architectures (see Figure 2).

### 2.3. Relevance Analysis Based on Class Activation Mapping for Semantic Segmentation

Deep learning provides the most effective approach to today’s intelligent systems; however, their prediction success is limited by the inability to explain human users’ decisions (interpretability); therefore, highlighting the most relevant features to discriminate between nerve and background could help visualize what is beneath the hood when using a neural network [51]. Here, a semantic segmentation-based extension of the gradient-class activation mapping (Grad-CAM++) algorithm [43] is used to provide an efficient data interpretability strategy, revealing fine-grained image details to capture the entire nerve completeness.

Let $S_{l}\left(\lambda \right)\in R^{R\times C}$ be a class-specific upsampled saliency map, regarding the output label λ∈{0,1}, as follows:(5)$$S_{l}\left(\lambda \right)=\left(\mu \circ \mathrm{ReLU}\right)\left(\sum_{d=1}^{D_{l}}\gamma_{l}^{d}\left(\lambda \right)F_{l}^{d}\right);$$
$F_{l}^{d}\in R^{R_{l}\times C_{l}}$ stands for the *d*-th feature map at layer *l* computed for a given input image I, $\mu :R^{R_{l}\times C_{l}}\to R^{R\times C}$ is an upsampling function, and $\gamma_{l}^{d}\left(\lambda \right)\in R^{+}$ is a saliency weight:(6)$$\gamma_{l}^{d}\left(\lambda \right)=1^{⊤}\left(A_{l}^{d}\left(\lambda \right)⊙\mathrm{ReLU}\left(\frac{\partial y(\lambda )}{\partial F_{l}^{d}}\right)\right)1,$$
where $A_{l}^{d}\left(\lambda \right)\in R^{R_{l}\times C_{l}}$, and:(7)$$y\left(\lambda \right)=E\left\{G_{ij}:\forall i,j|M_{ij}=\lambda \right\}$$
gathers a class-conditional score, being $G=W_{L}⊗F_{L-1}+B_{L}$, $G_{ij}\in G$, $M_{ij}\in M$, 1 is an all-ones column vector of proper size, and ⊙ stands for the Hadamard product [44].

Following the algorithm proposed by authors in [43], matrix $A_{l}^{d}\left(\lambda \right)$ in Equation (6) can be computed as:(8)$$A_{ij}^{d}\left(\lambda \right)=\frac{\frac{\partial^{2}y\left(\lambda \right)}{{\left(\partial A_{ij}^{d}\left(\lambda \right)\right)}^{2}}}{2\frac{\partial^{2}y\left(\lambda \right)}{{\left(\partial A_{ij}^{d}\left(\lambda \right)\right)}^{2}}+\sum_{m=1}^{R_{l}}\sum_{m^{\prime }=1}^{C_{l}}A_{mm^{\prime }}^{d}\left(\lambda \right)\frac{\partial^{3}y\left(\lambda \right)}{{\left(\partial A_{ij}^{d}\left(\lambda \right)\right)}^{3}}},$$
where $A_{ij}^{d}\in A_{l}^{d}\left(\lambda \right).$ It is worth mentioning that the weighted combination in Equation (6) favors our CAM-based approach to deal with different object orientations and views; meanwhile, the ReLU-based thresholding in Equations (5) and (6) constrained the relevance analysis to gather only positive gradients into $S_{l}\left(\lambda \right)$, indicating visual features that increase the output neuron’s activation rather than suppressing behaviors [52].

### 2.4. RFF-Based Semantic Segmentation Pipeline and Main Contributions

In summary, we propose a twofold deep learning pipeline from nerve structure segmentation. *(i)* An RFF-based layer (as discussed in Section 2.2) is coupled with three well-known shallow and complex architectures (as discussed in Section 2.1). To the best of our knowledge, this is the first attempt to combine kernel mappings within shallow and deep models to support nerve structure detection from 2D ultrasound images. *(ii)* A CAM-based extension for semantic segmentation (see Section 2.3) from RFF-based mappings is proposed to highlight the most relevant features (image regions) that favor discriminating between nerve and background. Figure 3 depicts our RFF-based semantic segmentation pipeline.

For concrete testing, we apply our RFF-based proposal within the FCN [12], U-net [14,41], and ResUnet [16] approaches. Our main aim is to improve the representation learning benefits of deep models using robust kernel mappings. Then, the RFF layer is used after the last pooling block in FCN (see Figure 1). Our kernel-based enhancement is located at the bottleneck block in U-net and ResUnet models (see Figure 2). Of note, the ResUnet architecture reformulates the U-net model through residual blocks (see Figure 4). In addition, we search the place with lower features number by RFF computational cost, $O(Q^{\prime }P^{\prime })$. Furthermore, we conducted various experiments that concluded that these are the better place for the RFF layer. These experiments were located the RFF layer in different places.

## 3. Experimental Setup

Our RFF-based deep learning enhancement approach was tested as a tool to support nerve structure segmentation from 2D ultrasound images. In particular, such a semantic segmentation was studied from three well-known deep learning architectures: U-net, ResUnet, and FCN (as exposed in Section 2). Thereby, we aimed to demonstrate the discriminative capability and interpretability benefits of our kernel-based improvement. In the following we present the studied datasets. Next, we describe the quantitive assessment, method comparison, and implementation details.

### 3.1. Ultrasound Image Datasets for Nerve Structure Segmentation

–*Nerve-UTP*: This dataset was acquired by the Universidad Tecnológica de Pereira (https://www.utp.edu.co, accessed on 17 November 2021) and the Santa Mónica Hospital, Dosquebradas, Colombia. It contains 691 images of the following nerve structures: the sciatic nerve (287 instances), the ulnar nerve (221 instances), the median nerve (41 instances), and the femoral nerve (70 instances). A SONOSITE Nano-Maxx device was used, fixing a 640×480 pixel resolution. Each image was labeled by an anesthesiologist from the Santa Mónica Hospital. As prepossessing, morphological operations such as dilation and erosion were applied. Next, we defined a region of interest by computing the bounding box around each nerve structure. As a result, we obtained images holding a maximum resolution of 360×279 pixels. Lastly, we applied a data augmentation scheme to obtain the following samples: 861 sciatic nerve images, 663 ulnar nerve images, 123 median nerve images, and 210 femoral nerve images (1857 input samples).–*Nerve segment dataset* (NSD): This dataset belongs to the Kaggle Competition repository [42]. It holds labeled ultrasound images of the neck concerning the brachial plexus (BP). In particular, 47 different subjects were studied, recording 119 to 580 images per subject (5635 as a whole) at 420×580 pixel resolution. For concrete testing, we performed a pruning procedure to remove images with inconsistent annotations as suggested by authors in [18,19,20], yielding to 2323 samples.

### 3.2. Method Comparison, Performance Measures, and Implementation Details

To compare the performance of our RFF-based framework for nerve structure segmentation that includes RFF-FCN, RFF-U-net, and RFF-ResUnet, where RFF stands for approximating kernel mapping (see Figure 3), we considered the following relevant state-of-the-art approaches: (i) FCN [12], (ii) U-net [14,41], and (iii) ResUnet [16]. Moreover, for the NSD dataset, the following approaches are studied: (iv) an automatic nerve structure segmentation methodology founded on U-net algorithm [18], (v) an approach that couples linear Gabor binary patterns and deep learning [19], and (vi) an algorithm that comprises median filtering and dense atrous convolution with residual multi-kernel pooling to enhance a ResUnet strategy [20].

A hold-out cross-validation scheme is applied for all provided datasets, setting 70% of the samples for training, 10% for validation, and 20% for testing. Furthermore, as quantitative assessment concerning the semantic segmentation performance, the sensitivity (Sen), the specificity (Spe), the Dice coefficient, the intersection over union (IOU), the area under the ROC curve (AUC), and the geometric mean (GM) are reported on the testing set, which can be written as follows: (9)$$\begin{array}{cc} Sen[\%]\;=\; & 100\frac{TP}{TP+FN} \end{array}$$(10)$$\begin{array}{cc} Spe[\%]\;=\; & 100\frac{TN}{TN+FP} \end{array}$$(11)$$\begin{array}{cc} Dice[\%]\;=\; & 100\frac{2TP}{2TP+FN+FP} \end{array}$$(12)$$\begin{array}{cc} IOU[\%]\;=\; & 100\frac{TP}{TP+FN+FP} \end{array}$$(13)$$\begin{array}{cc} GM[\%]=\; & \sqrt{Sen\times Spe}; \end{array}$$
where TP, FN, and FP represent the true positives, false negatives, and false positives predictions after comparing the actual and estimated label masks $M_{n}$ and ${\hat{M}}_{n}$ for a given input image $I_{n}$. The AUC can be computed by varying the decision boundary concerning the Sen and Spe measures [53].

Next, to measure the data interpretability quality (relevance analysis), two explanation map-based measures are introduced founded on the work proposed by authors in [43]. Thereby, let $\tilde{I}\left(\lambda \right)=\tilde{S}\left(\lambda \right)⊙I$ be the explanation map of image I with respect to the normalized class activation mapping $\tilde{S}\left(\lambda \right)=S\left(\lambda \right)/\max\left(S\left(\lambda \right)\right)$ at a given layer of interest. Moreover, let $\tilde{y}\left(\lambda \right)=E\left\{{\tilde{G}}_{ij}:\forall i,j|M_{ij}=\lambda \right\}$ be the expected class-conditional score concerning $\tilde{S}\left(\lambda \right)$, where $\tilde{G}=W_{L}⊗{\tilde{F}}_{L-1}+B_{L}$, ${\tilde{G}}_{ij}\in \tilde{G}$, fixing ${\tilde{F}}_{0}=\tilde{I}\left(\lambda \right),$ that is, the explanation map $\tilde{I}\left(\lambda \right)$ feeds the deep learning predictor in Equation (1) till the penultimate layer that holds a linear activation to preserve a class-conditional score activation as in Equation (7). Then, the following relevance analysis measures arises:(14)$$\begin{array}{cc} Increase\;Confidence= & \sum_{n=1}^{N}\frac{\vartheta ({\tilde{y}}_{n}\left(\lambda \right)>y_{n}\left(\lambda \right))}{N}100\left[\%\right] \end{array}$$(15)$$\begin{array}{cc} Win(M_{r},M_{r^{\prime }})= & \sum_{n=1}^{N}\frac{\vartheta ({\tilde{y}}_{n}\left(\lambda |M_{r}\right)>{\tilde{y}}_{n}\left(\lambda |M_{r^{\prime }}\right))}{N}100\left[\%\right] \end{array}$$
where ϑ(·) is an indicator function that returns 1 when the argument is true. For the *Increase Confidence* measure, the ideal value equals 100[%] and aims to quantify how an explanation map highlights the most relevant regions for decision-making, e.g., accurate nerve segmentation. Namely, it counts, as a percentage value, the number of images enhanced by the explanatory map that provides only the image-relevant patterns instead of the whole image to increase the prediction score. Then, for pair-wise comparison, the *Win* approach computes the percentage of times in which the explained map’s confidence of a model $M_{r}$ is better than in model $M_{r^{\prime }}$. For FCN, U-net, ResUnet, RFF-FCN, RFF-U-net, and RFF-ResUnet an Adam optimizer is fixed, using a $10^{-3}$ learning rate value in the Nerve-UTP dataset and $10^{-4}$ for NSD. A Dice-based loss is employed in Equation (2), as follows:(16)$$L_{Dice}\left(M_{n},{\hat{M}}_{n}\right)=2\frac{1^{⊤}\left(M_{n}⊙{\hat{M}}_{n}\right)1+\epsilon }{1^{⊤}M_{n}1+1^{⊤}{\hat{M}}_{n}1+\epsilon },$$
where ϵ=1 avoids numerical instability. A batch size of 32 samples is fixed and the *Q* hyper-parameter in Equation (4) is empirically fixed to preserve a feature map showing a squareform (image shaped) as $8\times 8\times \overset{˘}{Q}$, where the factor $\overset{˘}{Q}$ is serached within the set {8,16,32,64,128}.

All experiments were carried out in Python 3.8, with the Tensorflow 2.4.1 API, on a Google Colaboratory environment (code repository: https://github.com/cralji/RFF-Nerve-UTP, accessed on 17 Noember 2021).

## 4. Results and Discussion

### 4.1. Semantic Segmentation Results

Figure 5 depicts a representative example of the 2D nerve segmentation process for both Nerve-UTP and NSD datasets (FCN, U-net, ResUnet, RFF-FCN, RFF-U-net, and RFF-ResUnet results are shown). The $\overset{˘}{Q}$ hyperparameter for RFF-based methods is fixed as 128, 64, and 8 for RFF-FCN, RFF-Unet, and RFF-ResUnet on Nerve-UTP, and 128, 8, and 8 on NSD. Overall, our kernel mapping approach enhances the segmentation precision, avoiding, in most cases, false-positive prediction.

Concerning the Nerve-UTP dataset, the sciatic and femoral nerves provide the most challenging scenarios. Indeed, straightforward models such as FCN, U-net, and ResUnet cannot capture boundary regions of the nerve. However, after our RFF-based improvement, the nerve localization is more accurate. Consecutively, the BP in NSD exhibits a more difficult task compared to Nerve-UTP. As seen, the input image gathers noisy samples, which can be related to attenuation, artifacts, and speckle noise [5]. Again, FCN and U-net algorithms show poor performances, e.g., false-positive regions are highlighted as nerve; however, their RFF-based alternatives mitigate such false-positive predictions. It is worth mentioning that both ResUnet and RFF-ResUnet provide false-positive segmentation, which can be explained by the overfitting issue of deeper architectures [33].

Table 1 presents the comparison results between straightforward state-of-the-art methods for semantic segmentation and our RFF-based enhancement concerning the Sen, Spe, Dice, IOU, and GM quantitative assessments (see Equations (9) to (13)). In addition, a non-parametric Friedman test was computed for statistical significance. The null hypothesis was that all algorithms perform equally [54,55]. For concrete testing, we fixed the significance threshold as *p*-value < 0.05. In this sense, a Chi-square of 2856.32 was obtained for the Dice measure (*p*-value $=1.75\;\times \;10^{-218}$). Of note, all remaining measures also reject the null hypothesis.

At a glance, our RFF-based enhancement favors the segmentation prediction in most cases; especially, FCN and U-net architectures are favored by our kernel mapping to find a representative feature space to discriminate between nerve and background. Indeed, RFF-FCN and RFF-Unet obtain the first ranking places for most of the studied measures. However, again, as shown in Figure 5, ResUnet and RFF-Unet suffer from overfitting, i.e., see the low Sen values of the ResUnet method for challenging nerves. Still, RFF-ResUnet aims to prevent such behavior by enhancing the segmentation assessment for the BP identification task. Remarkably, the RFF-FCN and RFF-Unet achieve the best ranking (first and second place) for Dice and IOU, which are often used to test semantic segmentation tasks. Then, our kernel approach allows preserving a trade-off between network complexity and representation learning capability [34,36].

Next, we applied a pair-wise post hoc analysis regarding the Dice results reported in Table 1, to compute a *p*-value after the statistical comparison between models $M_{r}$ and $M_{r^{\prime }}$ [55]. As seen in Table 2, FCN vs. RFF-FCN, U-net vs. RFF-Unet depict *p*-value < 0.05, that is, our kernel-based approach allows obtaining better segmentation results holding a pair-wise statistical significance for shallow architectures. ResUnet and RFF-Unet present a similar performance (*p*-value > 0.05). In this sense, our kernel layer concedes similar segmentations to residual blocks coupled with a U-net scheme. Similarly, RFF-FCN vs. RFF-U-net displays a *p*-value =0.076, which shows that akin detections are retrieved after adding an RFF layer to FCN and U-net.

Lastly, Table 3 exhibits the method comparison results for the Dice measure on 2D brachial plexus nerve segmentation (NSD as a well-known Kaggle Competition dataset). At first sight, our RFF-Unet brings the best nerve segmentation results, improving at ∼3[%] its straightforward strategy U-net. The kernel improvements of FCN and ResUnet also afford Dice boosting. Though state-of-the-art methods, such as those of [19,20], introduce some preprocessing or feature extraction techniques before the deep-learning-based prediction, U-net-based methods (as the one proposed by authors in [18]) seem to be more appropriate for NSD, which poses a challenge concerning the noisy images, requiring a suitable representation learning.

### 4.2. Relevance Analysis Results

Figure 6 and Figure 7 show some visual inspection results for the Nerve-UTP and NSD datasets. In particular, the normalized class activation mapping ${\tilde{S}}_{l^{\prime }}\left(\lambda \right)$ is plotted as a heatmap on the 2D input image I (some illustrative examples of the testing set are selected), where $l^{\prime }$ is the convolutional layer just before our RFF-based enhancement (λ=0 stands for background class and λ=1 for nerve). As seen, our kernel mapping helps focalize the class activation maps on image regions related to the nerve structure.

Notably, our RFF-based improvement prunes the representation learning stage by avoiding nerve pixels when studying the background class; namely, our approach benefits the network attention to the nerve structure from ultrasound images. Yet, for the NSD, this is not appreciable but is supported by the semantic segmentation and CAM-based performance measures.

As exposed in the semantic segmentation results, the sciatic and femoral nerves present the most challenging scenarios, and external pixels close to the nerve structure are highlighted as relevant according to the class activation heatmaps; however, ultrasound images provide non-stationary conditions (shift-variant patterns), being necessary to hold neighborhood pixels around the class of interest to promote a proper segmentation.

Further, we analyze the concentration of the class activation maps by averaging the target region of the testing samples (see Figure 8 and Figure 9). The median operator is used to avoid outliers. Then, centering and scaling are applied for visualization, and the average heatmap is shown concerning background and nerve labels. Notably, RFF enhancement applied to U-net architecture undergoes a close neighborhood-based representation to discriminate between background and nerve. Indeed, U-net’ class activation maps exhibit neither salient information for the nerve nor the background class. Regarding ResUnet and RFF-ResUnet, condensed class activation maps are obtained for the nerve class. Still, as presented in Table 1, overfitting issue arises for such architectures with lower-ranking performances compared to FCN and U-net variants. On the other hand, the U-net architecture highlights the nerve region and its surroundings almost equally, but our proposed RFF-U-net makes the nerve stand out, showing a better representation. In addition, for ResUnet and RFF-ResUnet, our proposal reduces the background attention of the network.

Next, Table 4 reports the quantitative results for relevance analysis using the explanation map-based measures in Equations (14) and (15). Note that the increased confidence aims to compute the score gain within a given model M after feeding the network with the explanation map $\tilde{I}\left(\lambda \right);$ meanwhile, the Win assessment compares the score gain of $M_{r}$ and $M_{r^{\prime }}$. Overall, nerve confidence boosts are achieved under easy semantic segmentation scenarios, e.g., ulnar and median. Conversely, for sciatic and femoral, the increased confidence is low. Nonetheless, our kernel approach hikes the network score from explanation maps for shallow architectures such as FCN and U-net. Lastly, the Win-based measure results prove that our RFF-based variants (RFF-FCN, RFF-Unet, and RFF-ResUnet) improve the model score from their explanation maps, pushing relevant and discriminant input image patterns.

## 5. Conclusions

We proposed a kernel-based enhancement to support 2D nerve structure segmentation from ultrasound images using deep learning. Our proposal incorporates a random Fourier features (RFF)-based layer for kernel mapping, e.g., a Gaussian function [34], estimation within three well-known architectures for semantic segmentation: fully convolutional network (FCN) [12], U-net [14,41], and residual U-net (ResUnet) [16].

Our strategy seeks to improve the deep learning potential from the generalization capability of kernel methods, preserving mini-batch gradient descend optimization. Concretely, we apply an RFF-layer after the bottleneck end for U-net and ResUnet. Likewise, the RFF is joined after the last pooling in FCN. Furthermore, a class activation mapping algorithm, termed GradCam++ [43], is extended for semantic segmentation to visualize heatmaps that reveal the model’s ability to extract relevant features from ultrasound images. Next, explanation maps are used as quantitative assessment concerning the increased confidence (deep learning score before the last decision-making layer) for semantic segmentation tasks. To the best of our knowledge, this is the first attempt to enhance 2D ultrasound image segmentation for nerve structure identification using both shallow and deep networks while preserving class activation interpretability.

Experiments were carried out on two ultrasound 2D image datasets: (i) Nerve-UTP that belongs to the Universidad Tecnológica de Pereira and the Santa Mónica Hospital, Dosquebradas, Colombia; holding ultrasound images of sciatic, ulnar, median, and femoral nerves. (ii) NSD as a Kaggle Competition dataset [42], gathering ultrasonography records of the brachial plexus.

Obtained results prove that our RFF-based improvement facilitates the discrimination between nerve structure and background in terms of conventional performance measures: sensitivity, specificity, Dice, intersection over union, area under ROC curve, and geometric mean. Indeed, our approach can improve the discrimination effectiveness of straightforward (shallow) architectures, i.e., FCN and U-net, leveraging nonlinear kernel-based mapping within a deep learning paradigm; it preserves the performance of deeper approaches such as ResUnet, which holds a residual learning philosophy with more training parameters than FCN and U-net. In turn, the RFF-based mapping also favors the explanatory capacity of the segmentation algorithm, finding relevant maps that highlight salient image regions related to the nerve structure. All experiments were conducted on Python (TensorFlow and Keras), and both datasets and code are publicly available.

As future work, the authors plan to couple attention mechanisms for semantic segmentation [56] within the introduced kernel-based representation. Furthermore, an RFF-layer extension for direct image-based convolution operation could benefit the algorithm training and hyperparameter tunning [31]. Further, variational autoencoders can be incorporated within our scheme to avoid overfitting and to benefit the data interpretability [57]. Lastly, the extension of our deep learning pipeline to provide 3D nerve segmentation is an exciting research line [58]. 

## Figures and Tables

**Figure 1 sensors-21-07741-f001:**
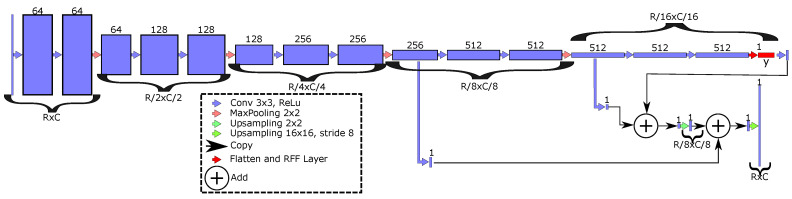
RFF-FCN architecture for semantic segmentation. The RFF layer is added after the last pooling (red block), number of filters, kernel size, stride, and nonlinear activation function are shown.

**Figure 2 sensors-21-07741-f002:**
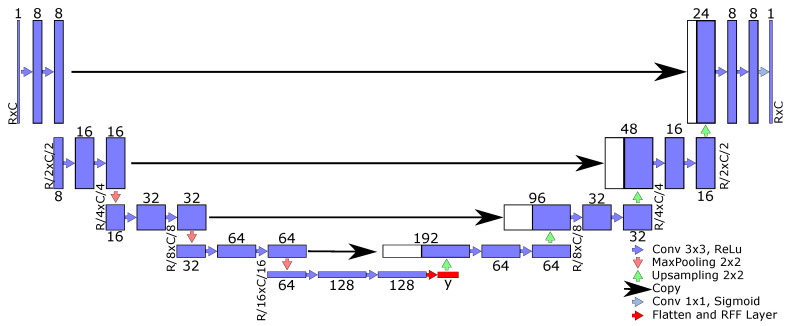
RFF-U-net architecture for semantic segmentation. U-shaped network to preserve both low and high-level features using an encoder–decoder approach. The RFF layer is located at the bottleneck (red block). The number of filters, kernel size, stride, and nonlinear activation function are depicted.

**Figure 3 sensors-21-07741-f003:**
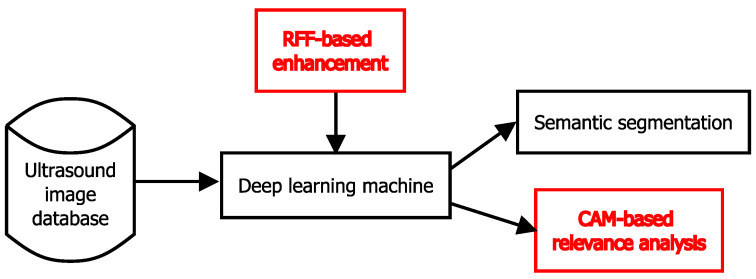
Nerve structure segmentation 2D from ultrasound images pipeline. FCN, U-net, and ResUnet architectures (see Section 2.1) are enhanced using an RFF-based layer (see Section 2.2). The predicted mask (nerve vs. background) and a class activation mapping (CAM)-based image (see Section 2.3) can be obtained as outputs.

**Figure 4 sensors-21-07741-f004:**
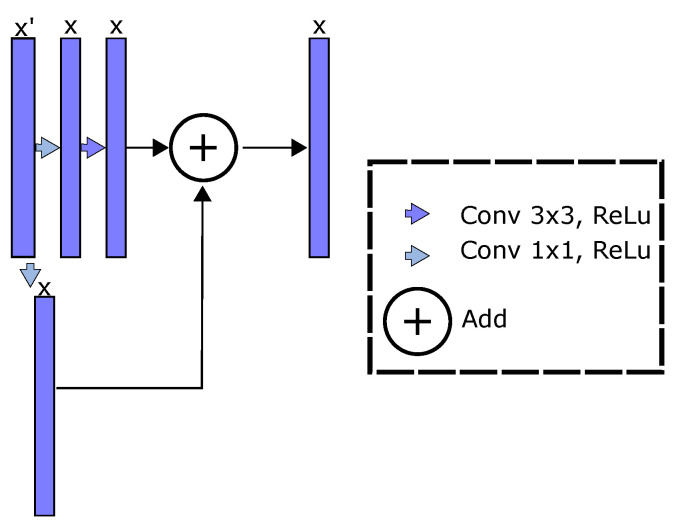
Residual block scheme for ResUnet-based semantic segmentation. The U-net approach is enhanced through residual blocks. The filter size, nonlinear activation, and architecture are presented.

**Figure 5 sensors-21-07741-f005:**
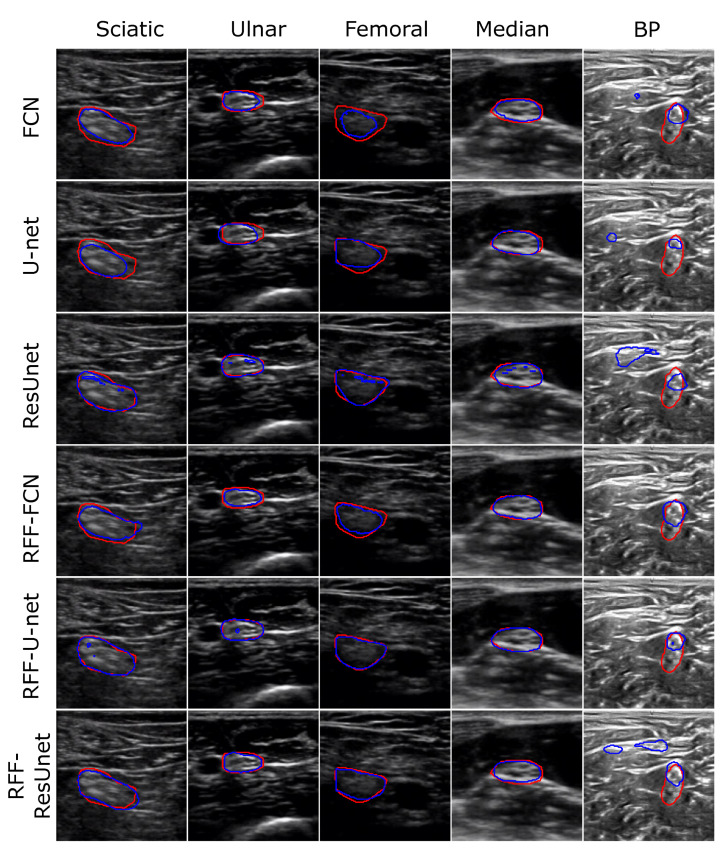
Visual inspection results for nerve structure segmentation (illustrative examples are shown). Red contour: target segmentation. Blue contour: predicted segmentation. The sciatic, ulnar, femoral, and median nerves are shown for the Nerve-UTP dataset. The brachial plexus (BP) of the NSD dataset is also presented. FCN [12], U-net [14,41], and ResUnet [16] algorithms were tested. Moreover, their RFF-based improvements (our proposal) are displayed, fixing the $\overset{˘}{Q}$ factor value as 128, 64, and 8 for RFF-FCN, RFF-Unet, and RFF-ResUnet on Nerve-UTP, and 128, 8, and 8 on NSD.

**Figure 6 sensors-21-07741-f006:**
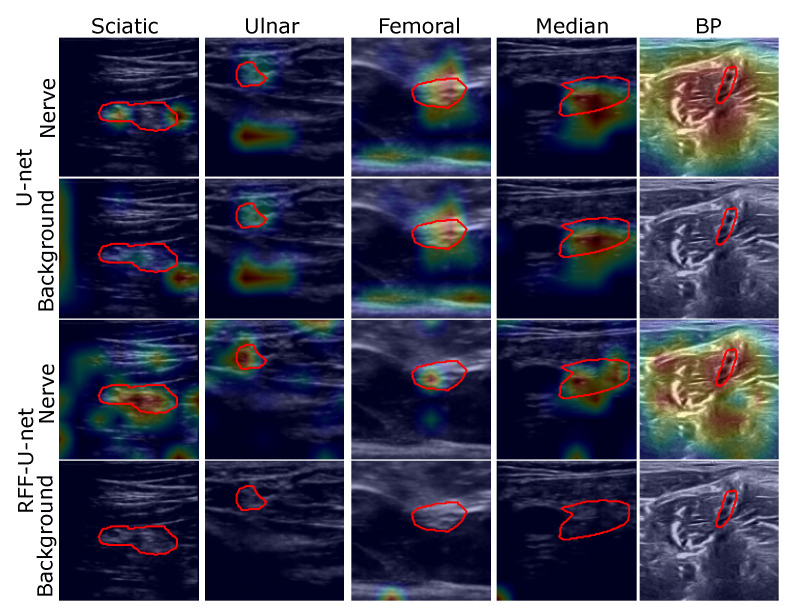
Unet and RFF-Unet relevance analysis results (illustrative examples are shown for the Nerve-UTP and NSD datasets). The class activation maps are shown in front of the input image (as an explanation map for visualization purposes)—red contour: target segmentation. Heatmaps are presented on a Jet colormap scale, where the blue color stands for low relevance, yellow for medium relevance, and red for high relevance.

**Figure 7 sensors-21-07741-f007:**
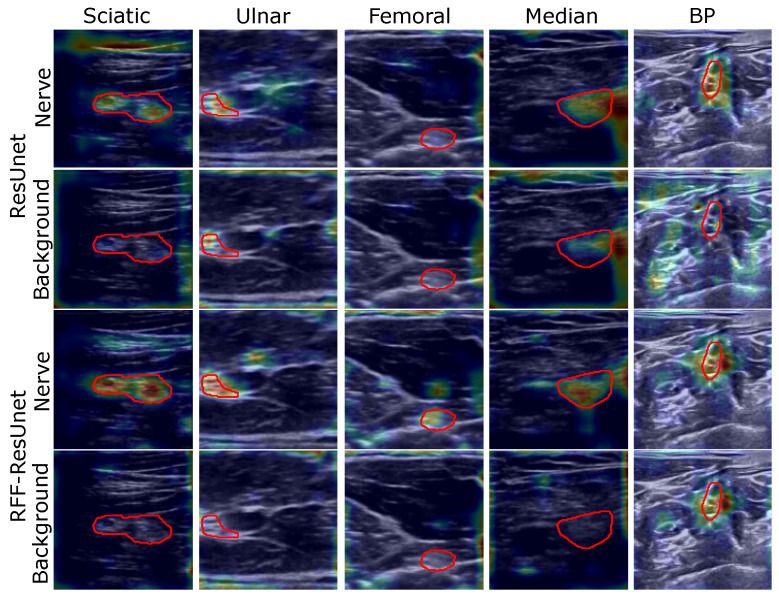
ResUnet and RFF-ResUnet relevance analysis results (illustrative examples are shown for the Nerve-UTP and NSD datasets). The class activation maps are shown in front of the input image (as an explanation map for visualization purposes)—red contour: target segmentation. Heatmaps are presented on a Jet colormap scale, where the blue color stands for low relevance, yellow for medium relevance, and red for high relevance.

**Figure 8 sensors-21-07741-f008:**
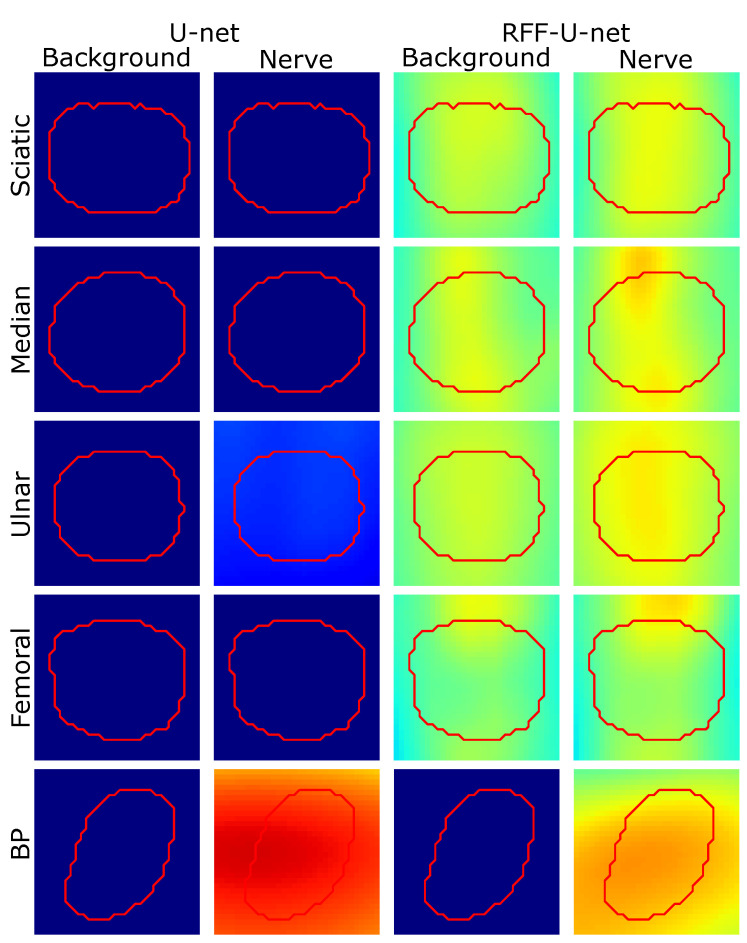
Region of interest-based relevance analysis results for Unet and RFF-Unet approaches (Nerve-UTP and NSD datasets). Red contour: median target segmentation region (rescaled and centered for visualization) along with the testing set instances. Heatmaps are presented on a Jet colormap scale, where the blue color stands for low relevance, yellow for medium relevance, and red for high relevance.

**Figure 9 sensors-21-07741-f009:**
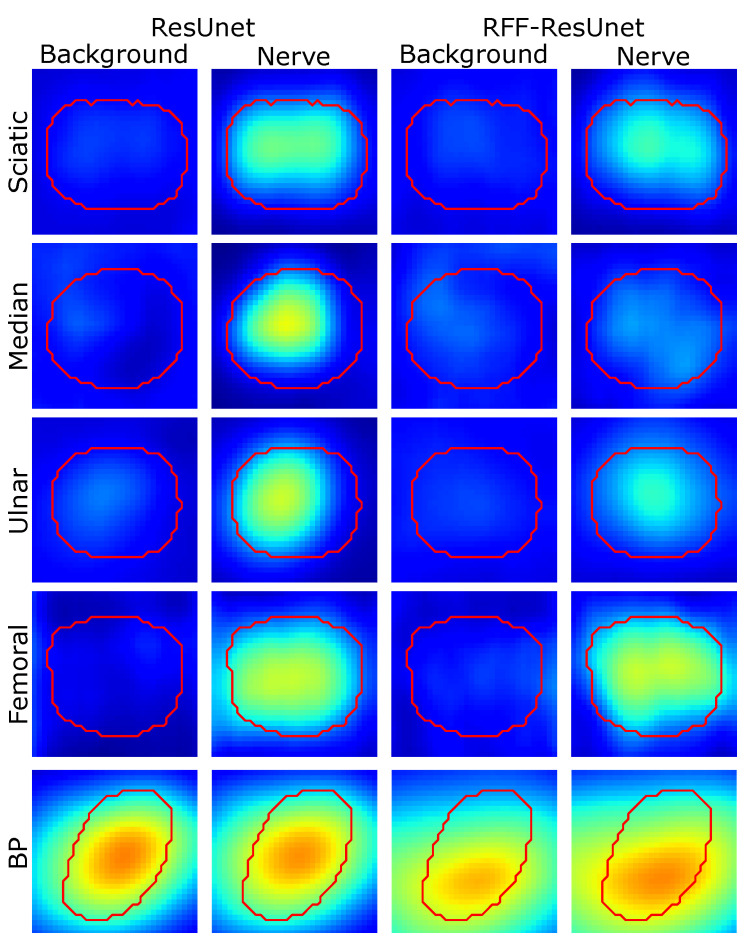
Region of interest-based relevance analysis results for ResUnet and RFF-ResUnet approaches (Nerve-UTP and NSD datasets). Red contour: median target segmentation region (rescaled and centered for visualization) along with the testing set instances. Heatmaps are presented on a Jet colormap scale, where the blue color stands for low relevance, yellow for medium relevance, and red for high relevance.

**Table 1 sensors-21-07741-t001:** Semantic segmentation results for nerve structure identification (the average along the testing set is reported for all provided measures). Bold: Highest performance among methods. Ranking stands for the average ranking from a Friedman statistical test. The sciatic, ulnar, femoral, and median nerve segmentation results are shown for the Nerve-UTP dataset. The brachial plexus (BP) results of the NSD dataset are also presented. FCN, U-net, ResUnet, and RFF-based enhancements (our approach) are presented. $\overset{˘}{Q}$ factor value equals 128, 64, and 8 for RFF-FCN, RFF-Unet, and RFF-ResUnet on Nerve-UTP’ nerves, and 128, 8, and 8 on NSD’s BP structure.

Model	Measure	Sciatic	Ulnar	Femoral	Median	BP	Ranking
FCN [12]	Sen [%]	72.1±8.9	60.8±26.8	66.0±22.8	66.5±23.4	75.13±20.68	5.2
	Spe [%]	99.8±0.4	99.9±0.3	99.9±0.2	100.0±0.1	99.54±0.43	1.8
	AUC [%]	99.6±0.8	96.1±13.0	97.4±9.1	96.9±12.5	97.91±6.94	2.2
	GM [%]	84.6±5.4	72.6±28.5	77.9±23.0	78.3±22.5	84.78±17.12	5.4
	Dice [%]	82.5±6.4	69.8±28.4	75.9±24.1	76.2±24.2	76.82±18.58	4.4
	IOU [%]	70.8±8.7	59.3±25.8	65.6±22.6	66.0±23.0	65.27±19.15	4.6
U-net [14,41]	Sen [%]	78.1±18.5	54.5±37.8	76.1±23.7	71.0±23.6	69.26±21.69	4.2
	Spe [%]	98.4±1.4	99.4±0.8	97.3±2.3	99.1±1.1	99.68±0.33	5.0
	AUC [%]	88.2±9.1	76.9±18.7	86.7±11.5	85.0±11.7	98.45±3.22	4.8
	GM [%]	86.5±13.8	62.0±39.4	82.6±23.7	80.2±24.5	98.45±3.22	4.2
	Dice [%]	76.9±15.2	54.7±36.3	73.6±22.7	73.1±23.6	74.32±19.45	5.6
	IOU [%]	64.5±16.8	45.8±32.2	62.0±21.0	61.7±22.0	62.18±19.48	5.6
ResUnet [16]	Sen [%]	84.7±8.4	68.5±29.9	75.4±19.0	73.0±32.0	63.52±19.55	3.0
	Spe [%]	99.3±0.7	99.7±0.5	99.2±1.2	99.7±0.4	99.46±0.47	4.6
	AUC [%]	92.0±4.1	84.1±14.9	87.3±9.4	86.3±15.9	96.29±6.60	4.6
	GM [%]	91.6±4.7	77.1±29.8	84.2±19.5	80.0±29.5	77.32±18.49	3.8
	Dice [%]	86.8±6.3	71.9±29.8	80.1±20.4	75.9±30.1	67.87±19.11	4.2
	IOU [%]	77.3±8.9	62.7±28.2	70.1±19.2	68.3±29.6	54.00±17.91	3.8
RFF-FCN	Sen [%]	80.8±8.4	67.9±21.9	74.4±13.7	80.3±18.4	75.48±18.99	3.0
	Spe [%]	99.7±0.3	99.9±0.3	99.7±0.7	99.9±0.1	99.55±0.41	2.0
	AUC [%]	99.5±1.2	98.0±8.6	99.1±3.6	97.6±11.2	98.92±2.82	1.2
	GM [%]	89.6±4.8	79.7±20.7	85.5±10.9	87.6±18.5	85.59±13.71	2.4
	Dice [%]	87.4±5.8	76.0±22.1	83.2±13.6	86.0±18.3	77.43±15.43	1.4
	IOU [%]	78.0±8.4	65.2±21.9	72.9±13.8	78.4±17.7	65.35±17.08	2.0
RFF-U-net	Sen [%]	83.8±10.9	71.5±25.2	73.6±18.8	77.0±22.6	79.24±21.67	2.6
	Spe [%]	99.4±0.7	99.8±0.2	99.4±1.0	99.7±0.3	99.40±0.47	3.6
	AUC [%]	91.6±5.6	85.7±12.6	86.5±9.5	88.4±11.4	98.14±4.05	3.8
	GM [%]	91.0±6.8	80.6±25.2	84.0±16.5	86.1±16.5	87.01±17.48	2.2
	Dice [%]	87.4±9.7	77.5±25.4	80.2±19.3	82.9±20.3	77.26±18.56	2.0
	IOU [%]	78.7±12.0	68.5±24.3	70.0±19.4	74.7±22.3	65.86±19.24	1.8
RFF-ResUnet	Sen [%]	84.4±10.1	71.5±28.4	72.4±15.6	83.8±11.9	60.48±19.50	3.0
	Spe [%]	99.2±0.7	99.7±0.4	99.3±1.1	99.7±0.3	99.67±0.38	4.0
	AUC [%]	91.8±5.0	85.6±14.1	85.9±07.9	91.8±6.0	94.72±7.83	4.4
	GM [%]	91.3±5.9	79.5±28.5	84.2±10.4	91.1±7.2	75.57±17.86	3.0
	Dice [%]	86.6±7.3	74.8±28.2	80.0±14.6	87.9±8.9	68.52±18.81	3.2
	IOU [%]	77.1±10.0	65.7±26.7	68.6±15.8	79.4±11.6	54.76±18.18	3.2

**Table 2 sensors-21-07741-t002:** Pair-wise post hoc analysis using the Friedman test concerning the Dice-based results in Table 1. The $r,r^{\prime }$ element reports the *p*-value after the statistical comparison between models $M_{r}$ and $M_{r^{\prime }}$. If *p*-value < 0.05, then statistical difference between approaches is accepted.

Method	FCN [12]	Unet [14,41]	ResUnet [16]	RFF-FCN	RFF-U-net	RFF-ResUnet
FCN [12]	−	0.001	0.001	0.001	0.001	0.001
U-net [14,41]	0.001	−	0.001	0.001	0.001	0.001
ResUnet [16]	0.001	0.001	−	0.002	0.853	0.001
RFF-FCN	0.001	0.001	0.002	−	0.076	0.001
RFF-U-net	0.001	0.001	0.853	0.076	−	0.001
RFF-ResUnet	0.001	0.001	0.001	0.001	0.001	−

**Table 3 sensors-21-07741-t003:** Method comparison results for NSD dataset (brachial plexus nerve segmentation). The average Dice is presented regarding the testing set. Bold: highest Dice-based performance. RFF-FCN, RFF-Unet, and RFF-ResUnet stand for our kernel-based deep learning enhancement.

Method	Dice [%]
Baby and Jereesh [18]	71.0
Kakade and Dumbali [19]	68.8
Wang et al. [20]	70.9
FCN [12]	64.9
U-net [14,41]	74.3
ResUnet [16]	67.9
RFF-FCN	62.9
RFF-U-net	77.3
RFF-ResUnet	68.5

**Table 4 sensors-21-07741-t004:** Relevance analysis results based on class-conditional score measures (the increase confidence and Win-based assessments are reported for Nerve-UTP and NSD datasets). The sciatic, ulnar, femoral, and median nerve segmentation results for Nerve-UTP, and the brachial plexus (BP) for NSD are considered. Win(M,RFF-M) is shown, where Win(RFF-M,M) = 100-Win(M,RFF-M).

Method	Relevance Measure	Sciatic	Ulnar	Median	Femoral	BP
FCN [12]	Increase Confidence [%]	0.0	6.8	4.0	2.4	100.0
	Win(FCN, RFF-FCN) [%]	43.6	53.4	32.0	42.9	51.0
U-net [14,41]	Increase Confidence [%]	1.7	30.8	4.0	2.4	0.0
	Win(U-net, RFF-Unet) [%]	0.0	6.0	0.0	0.0	95.7
ResUnet [16]	Increase Confidence [%]	0.0	8.3	8.0	0.0	3.4
	Win(ResUnet, RFF-ResUnet) [%]	39.5	32.3	48.0	21.4	51.0
RFF-FCN	Increase Confidence [%]	4.7	10.5	4.0	2.4	100.0
	Win(RFF-FCN, FCN) [%]	56.4	46.6	68.0	57.1	49.0
RFF-Unet	Increase Confidence [%]	0.0	3.0	0.0	3.0	4.3
	Win(RFF-Unet, U-net) [%]	100.0	94.0	100.0	100.0	4.3
RFF-ResUnet	Increase Confidence [%]	0.0	7.5	0.0	0.0	4.9
	Win(RFF-ResUnet, ResUnet) [%]	60.5	67.7	52.0	78.6	49.0

## Data Availability

Datasets are publicly available at: https://www.kaggle.com/c/ultrasound-nerve-segmentation (accessed on 5 October 2021), and https://www.kaggle.com/craljimenez/nerveutp (accessed on 5 October 2021).

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
