# Peer review of "Random Fourier Features-Based Deep Learning Improvement with Class Activation Interpretability for Nerve Structure Segmentation"

_sensors, 2021, doi:10.3390/s21227741_

Round 1
Reviewer 1 Report
The paper presents a kernel-based deep learning enhancement for nerve
structure segmentation. In a nutshell, a random Fourier features-based approach is utilized to complement three well-known semantic segmentation architectures, e.g., fully convolutional network, U-net, and ResUnet. Two ultrasound image databases for PNB are tested. Obtained results show that the kernel-based approach provides a better generalization capability from image segmentation-based assessments on different nerve structures. Besides, for data interpretability, a semantic segmentation extension of the GradCam++ for class-activation mapping is used to reveal relevant learned features separating between nerve and background.
The presented proposal favors both straightforward (shallow) and complex architectures (deeper neural networks).
The work is well organized, with current bibliography.
The work is of interest, coming in support of the medical field.
The scientific content is at a high level, the experimental results being presented correctly and clearly. Improvements in the field are highlighted by comparisons with other methods, on two data sets.
I recommend accepting the article.
Reviewer 2 Report
The paper suffers from the following:
- The English is too poor. The writing is so difficulty to read, as such, native English speaker is suggested to prove-read before submission.
- The proposed method is not clearly explained. The authors are suggested to clearly and simply explain the proposed method.
- The paper suffers from novelty. No significant novelty is found in the work. The small addition to the well-known deep learning models, contribute no significant knowledge.
- The volume of the segmented target is also required to be added in the evaluation process.
Reviewer 3 Report
The manuscript "Random Fourier features-based deep learning improvement with class-activation interpretability for nerve structure segmentation" proposes a kernel-based deep learning enhancement for nerve structure segmentation. They have utilized a random Fourier features-based approach to complement three semantic segmentation architectures, namely, CNN, U-net, and ResUnet. Their results show that the kernel-based approach provides a better generalization capability from image segmentation-based assessments on different nerve structures. I would recommend the authors to address the following comments before moving forward: 1- The language (English) of the manuscript needs significant improvement. There are a number of grammatical errors, typos, and long convoluted sentences that are difficult to parse. I think this is the main issue with the manuscript. 2- Some of the paragraphs are quite long. I think their content can be broken down to multiple paragraphs instead. For instance, in the Intro section. 3- I think too much mathematical details are provided in the manuscript. Some of these details can be moved to an appendix or a supplementary document file to avoid distracting the readers from the main context. For instance, in sections 2.3 or 2.2. 4- Some important details regarding the DL is missing from the manuscript. For instance, what architectures have been used for different neural networks? Which hyperparameters/optimizers? What process has been used to tune the hyperparameters? How long training/prediction process takes and on what computational resources?Author Response
Please see the attachment

Round 2
Reviewer 2 Report
The paper is well revised.
Reviewer 3 Report
The authors have addressed my comments. It can be published in the present format.